# Improving Post Training Neural Quantization: Layer-wise Calibration and Integer Programming

## Abstract

Lately, post-training quantization methods have gained considerable attention, as they are simple to use, and require only a small unlabeled calibration set. This small dataset cannot be used to fine-tune the model without significant over-fitting. Instead, these methods only use the calibration set to set the activations' dynamic ranges. However, such methods always resulted in significant accuracy degradation, when used below 8-bits (except on small datasets). Here we aim to break the 8-bit barrier. To this end, we minimize the quantization errors of each layer separately by optimizing its parameters over the calibration set. We empirically demonstrate that this approach is: (1) much less susceptible to over-fitting than the standard fine-tuning approaches, and can be used even on a very small calibration set; and (2) more powerful than previous methods, which only set the activations' dynamic ranges. Furthermore, we demonstrate how to optimally allocate the bit-widths for each layer, while constraining accuracy degradation or model compression by proposing a novel integer programming formulation. Finally, we suggest model global statistics tuning, to correct biases introduced during quantization. Together, these methods yield state-of-the-art results for both vision and text models. For instance, on ResNet50, we obtain less than 1% accuracy degradation — with 4-bit weights and activations in all layers, but the smallest two. Our code is publicly available at *https://github.com/papers-submission/CalibTIP*

## 1 Introduction

The pursuit of advanced Deep Neural Networks (DNNs) causes researchers to construct deeper and wider networks, making them expensive to use in terms of power and time. This increases the need for efficient implementations of these networks. Efficient networks reduce cloud-vendor costs and make it possible to run them on low-power devices such as smartphones and wearable devices. The most common off-the-shelf approach to improving network efficiency is quantization, which reduces the numerical precision of the network and its complexity and memory footprint.

DNN quantization techniques can be classified as either post-training or quantization-aware training (QAT) techniques (Han et al., 2015; Courbariaux et al., 2015; Hubara et al., 2017; Zhou et al., 2016). Although QAT techniques, in general, achieve better results, there are important real-world scenarios in which they are not applicable. These are the cases where the training data is sensitive or simply unavailable at the time of deployment. For instance, when off-the-shelf or legacy models are being used, or when medical records are involved. Therefore, much attention has recently been dedicated to post-training quantization methods (Nagel et al., 2019; Banner et al., 2018; Zhao et al., 2019), which can be more easily applied in practice. These methods allow for network quantization to happen seamlessly when deployed, without requiring additional information from the user except a small unlabeled calibration set.

Unfortunately, post-training quantization below 8-bit always incurs significant accuracy degradation and in some cases even higher numerical precision is required. In this paper, our goal is to break this barrier by distilling all the information the pre-trained model and calibration set encode. Our goal is to find an optimal scheme for current state of the art hardware which usually support 16,8,4 bits data types with per-channel quantization of the weights. To that end, we suggest a three-stage

pipeline that consists of methods applied solely on a small calibration set to reduce the local error introduced during the quantization process (e.g., round-off errors) followed by integer programming to determine the bit-width of different layers so that the overall accuracy degradation is minimized. Even without using mixed-precision, the suggested method is much less prone to over-fitting than current methods and yields best in class results for 8-bits Mobilenet-V2 and BERT-base trained on ImageNet and SQuAD1.1 datasets, respectively. Our paper suggests several contributions for mixed-precision post-training quantization:

1. **AdaQuant:** A layer-by-layer optimization method that minimizes the error between the quantized layer output and the full-precision layer output. This method can consume only a small calibration dataset from training data without overfitting. In a comprehensive study, we show that AdaQuant defines a new state-of-the-art for post-training quantization on several networks and tasks, including vision models (Resnet18, Resnet50, MobilenetV2) and language (BERT).

2. **Integer programming:** As some parts of the network may allow lower precision compared to other layers, we suggest an integer-linear programming based approach for determining the precision level of different layers. This method aims at maximizing either the expected speedup or savings in power consumption without violating a predefined constraint on network accuracy degradation or compression.

3. **Batch-norm tuning:** Following quantization we observe an inherent bias in the mean and the variance of batch norm statistics. We show that by employing the re-estimated statistics in batch normalization, much of the quantized network degradation can be recovered.

4. **Light and Advanced pipelines:** We analyze the advantages and disadvantages of each of the given methods and suggest two pipelines: (1) light pipeline that does not require a backward pass, thus can be invoked even on inference-only hardware; and (2) Advanced pipeline that includes also AdaQuant and bias tuning.

## 2 RELATED WORK

There has been a significant effort to accelerate inference via quantization (Courbariaux et al., 2015; Han et al., 2015; Rastegari et al., 2016; Zhou et al., 2017). These works involve re-training in order to compensate for the degradation due to the quantization process. Post-training quantization, on the other hand is applied to a model after it was trained. Thus, it avoids re-training and as such it is much simpler to use. However, naively quantizing a full-precision model to INT4 or lower to accelerate computation usually incurs significant accuracy degradation (Krishnamoorthi, 2018; Jacob et al., 2018).

**AdaQuant:** A recent post-training quantization method (Nagel et al., 2020), termed AdaRound, suggested optimizing the rounding policy. Instead of using the predominant rounding-to-nearest approach, they suggest formulating a per-layer quadratic optimization problem to optimize the round-off error. Our proposed method, AdaQuant, takes another step and relaxes AdaRound's implicit constraint which forces the quantized weights to be within $\pm 1$ of their round-to-nearest value. This is done by optimizing the weights and quantization parameters of each layer separately, over the calibration set, to minimize the MSE between the layer's original and quantized outputs. As oppose to AdaRound we apply AdaQuant to find optimal quantization not only to weights but also to activations. In addtion we suggest two flavors for AdaQuant: (1) **parallel-AdaQuant** suited for mixed precision setting; (b) **sequential-adaquant** which suited for fixed configuration.

**Integer programming:** Early work by Lin et al. (2016) used a convex optimization formulation which results in a simple greedy compression scheme. Aflalo et al. (2020) used a combinatorial optimization approach for network pruning. Their problem was formulated as a Knapsack problem that optimizes the trade-off between the channels importance and their associated computational cost. Cai et al. (2020) finds a mixed-precision configuration with a guaranteed Pareto efficient allocation with respect to model size and accuracy degradation. While this provides a "best-effort" standard (e.g., the configuration cannot be further compressed without hurting accuracy), it does not suggest which of all possible outcomes is best. To the best of our knowledge, this work is the first to formalize a generic integer program, which can easily be adapted to various types of models and requirements with a clear objective and constraints.

**Batch norm tuning:** Finkelstein et al. (2019) were the first to recognize that a significant source of degradation is a shift in the mean activation value. They show a simple method to compensate for this bias by updating the bias terms. Nagel et al. (2019) suggest to equalize the weight ranges in the network and correct biases in the error that are introduced during quantization .Recently Sun et al. (2019) suggested batch norm tuning for FP8 models. Here we detail how to perform this procedure on a per-channel quantized (PCQ) model with fused batch-norm layers. The procedure is light as it only requires to invoke the quantized model few times (on the calibration set) and adjust the quantization parameters.Moreover after retuning the BN layers can be reabsorbed which reduces the inference complexity. To the best of our knowledge we are the first to suggest it.

## 3 OPTIMIZING THE QUANTIZATION PIPELINE

In most post-training quantization settings, a model and a small unlabeled calibration set are given. To avoid overfitting the calibration set, most studies utilize it only to extract the network's internal statistics, which is later used to set the quantization parameters.

Here we suggest using the calibration set much more extensively to tune the model while avoiding over-fitting the data. In the following subsections, we detail three different optimization methods over the calibration set: (1) AdaQuant, a layerwise optimization of weights and quantization parameters; (2) an integer programming formulation for a mixed-precision setting; and (3) Batch Normalization Tuning (BNT), for tuning the model's internal statistics to match the numerical precision setting. We discuss the strengths and weaknesses of each method and suggest an optimization flow that exploits all the additive merits and leads to state-of-the-art results.

### 3.1 ADAQUANT - LAYERWISE OPTIMIZATION OVER THE CALIBRATION SET

Several researchers suggested per-tensor optimization to reduce quantization error by minimizing some form of MSE objective between the quantized and the full-precision tensor $X$ (either weights or activations). They look for an optimized quantization step size $\hat{\Delta}$ obtained by

$$\hat{\Delta} = \arg\min_{\Delta} ||X - Q_{\Delta}(X)||^2; \qquad Q_{\Delta}(X) = \Delta \cdot \left\lfloor \frac{X}{\Delta} \right\rceil, \tag{1}$$

where $Q(\cdot)$ is the quantization function. Although these methods are fast and easy to use, they often result in an inferior solution — the loss in eq. 1 is sub-optimal, as it penalizes all the quantization errors equally. However, the loss should penalize more quantization errors which affect the classification. Accordingly, researchers suggested *Quantization-Aware-Training* (QAT) methods to fix this error by training the entire model at once. However, those methods have three limitations: (a) they require the large training set to avoid over-fitting, (b) they approximate the back-propagation gradients through discrete function (the quantizer) and (c) they have high computational and memory footprints. We suggest a modified objective for per-layer joint optimization of the weights and quantization parameters.

$$\left(\hat{\Delta}_w, \hat{\Delta}_x, \hat{V}\right) = \arg\min_{\Delta_w, \Delta_x, V} ||WX - Q_{\Delta_w}(W + V) \cdot Q_{\Delta_x}(X)||^2, \tag{2}$$

where $V$ is a continuous variable added to $W$ and the quantized network weights are defined as $W_q = Q_{\hat{\Delta}_w}(W + \hat{V})$. In this new objective the quantized tensor is not required to be "close" to the original tensor, as in eq. 1, thus benefiting from the flexibility that Quantization-Aware-Training methods have. Yet, it can be executed in parallel over all layers and is much less prone to over-fitting. Moreover, under a fixed configuration we can optimize the model globally and infer the error between layers. Thus, instead of running AdaQuant on all layers in parallel we can run it sequentially and fix the error induced by quantaizing former layers. Thus, Eq. 2 changes to:

$$\left(\hat{\Delta}_{w_l}, \hat{\Delta}_{x_l}, \hat{V}_l\right) = \arg\min_{\Delta_{w_l}, \Delta_{x_l}, V_l} ||W_l X_l - Q_{\Delta_{w_l}}(W_l + V_l) \cdot Q_{\Delta_{x_l}}(X_l^q)||^2, \tag{3}$$

$$X_q = \sigma(Q_{\Delta_{w_{l-1}}}(W_{l-1} + V_{l-1}) \cdot Q_{\Delta_{x_l}}(X_{l-1}^q)) \tag{4}$$

where $\sigma(\cdot)$ is some activation function.

Note, that sequential AdaQuant should not be applied before the bit allocation was set as it optimize over noisy input obtain from predecessor quantized layers. We evaluate both flavors of adaquant

(named, *AdQuant* and *sequential-AdaQuant* and detail our finding in section 5.1. We note that AdaQuant also optimizes over biases and offsets and optimized fused conv-bn-relu layers when present; these were removed from the formulation in Equation 2 for simplicity.

**Size of calibration set**  Perhaps surprisingly, although we experiment with a very small calibration set, no over-fitting is observed. Let us examine a simple fully connected layer $W \in R^{M \times N}$. The input and output are of sizes $N$ and $M$, respectively. For each output we have $B$ equations and $N$ separate parameters (i.e., with no overlap in parameters between different outputs). Therefore if $B \ll N$ we generically have an infinite amount of solutions and we can overfit the data. If $B \gg N$ then we might underfit the data. Thus, the size of the calibration set required for AdaQuant should roughly be $O(N)$. A similar derivation for convolution layers reveals that the calibration size should have $B \geq \frac{C_i \cdot k^2}{HW}$ samples to avoid over-fitting, where $B$ is the number of unique samples, $k$ is the convolution's kernel size, $C_i$ and $C_o$ is the number of input and output channels respectively and $H, W$ represent height and width. In fig. 1 we compare AdaQuant to current state-of-the-art methods including QAT with knowledge distillation (QAT-KLD) (Kim et al., 2019) and

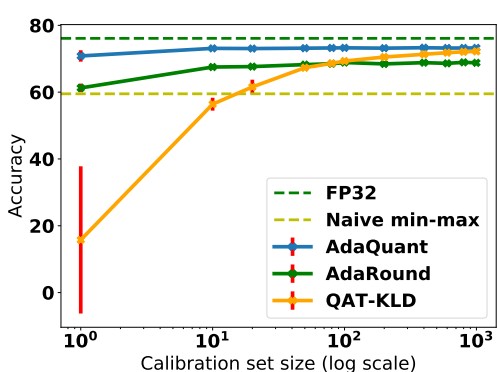

Figure 1: Comparison of different optimization methods over ResNet-50 quantized to 4 bit except the first and the last layers which were kept in 8bit. Even optimizing on a single image drastically improves the results but as expected have a high variance (red bar). The variance decreases rapidly as calibration set size increases.

AdaRound (Nagel et al., 2020). For each method, we measured the top-1 accuracy with respect to the number of samples in the calibration set over five runs and present the mean and standard deviation. As can be seen, AdaQuant is superior to previous methods and specifically excels on small calibration sets. Remarkably, AdaQuant does not overfit even when optimized on a single image. Additional details can be found in section A and D of the Appendix.

## 3.2 Per-layer bit allocations with integer programming

AdaQuant significantly enhances network accuracy at lower bit widths. However, it is often not sufficient by itself to attain acceptable accuracy. Therefore, in practical use cases, the user would like to balance between accuracy and performance (e.g., power and speed), by setting several layers to higher precision. Our high-level goal in this section would be to optimize the overall network performance while maintaining a predefined accuracy degradation or a model compression constraint.

In the following, we provide an integer-programming (IP) formulation for optimizing per-layer bit allocations. Depending on the needs, our performance metrics $\mathbb{P}$ would be either the execution time of the network or its power consumption. Also, with every layer quantization, there is an associated quantization error that affects the training loss $\mathcal{L}$. We chose the latter to be our penalty metric. Integer programming is applied in those situations where a given problem can clearly be represented in the form of a linear relationship between different decision variables. Unlike other previous works on compression, it attains a **global optimum**. For example, Lin et al. (2016) suggested a convex optimization problem, but the constraints and the objective are not linear. This typically has a drastic impact on convergence time, and the quality of the results since the Simplex method can no longer be applied (Van Doormaal & Raithby, 1984).

**Basic formulation**  We are given a neural network with $L$ layers. For each layer $l$, we have weights $W_l$ that need to be multiplied with activations of the previous layer $X_{l-1}$. Such lower bit width multiplications can be executed by quantizing the weights and activations to achieve higher throughput and energy-efficient solutions. Let $W_l^k$ and $X_{l-1}^n$ represent a quantized version of $W_l$ and $X_{l-1}$ to $k$ and $n$ bits, respectively. For each layer $i$, a low-bit width multiplication $W_l^k \cdot X_{l-1}^n$ results in a loss degradation $\Delta \mathcal{L}_l^{k,n}$ and in performance improvement $\Delta \mathbb{P}_l^{k,n}$ with respect to the original product

$W_l \cdot X_{l-1}$. This performance improvement measure needs to be additive and sum up to a total benefit in end-to-end network performance (e.g., power, model size, etc.). Our goal would be to maximize the total performance improvement without exceeding the total network degradation $\Delta \mathcal{L}$.

We now turn to solve the above problem using an integer program. We define a binary variable $I_l^{k,n}$, which is set to one if and only if the weights $W_l^k$ are multiplied with the activations $X_{l-1}^n$ at layer $l$; otherwise we set the indicator to zero i.e., $I_l^{k,n} = 0$. Then, the basic bit allocation problem can be formulated as follows:

$$\text{Maximize} \quad \sum_{l=0}^{L-1} \Delta \mathbb{P}_l \tag{5a}$$

$$\text{Subject to} \quad \sum_l \Delta \mathcal{L}_l \leq \Delta \mathcal{L}, \tag{5b}$$

$$\forall l \in \{1, ..., L\} : \Delta \mathbb{P}_l = \sum_{k,n} I_l^{k,n} \cdot \Delta \mathbb{P}_l^{k,n}, \Delta \mathcal{L}_l = \sum_{k,n} I_l^{k,n} \cdot \Delta \mathcal{L}_l^{k,n} \tag{5c}$$

$$\forall l \in \{1, ..., L\} : \sum_{k,n} I_l^{k,n} = 1, I_l^{k,n} \in \{0, 1\} \tag{5d}$$

The objective function (3a) maximizes the total performance improvement. Constraints (3b) and (3c) ensure that the total degradation in loss and the total improvements in performance due to the quantization of layer $l$ to k-bit-weights and n-bit-activations would be $\Delta \mathcal{L}_l$ and $\Delta \mathbb{P}_l$, respectively. Equation (3d) states that the restriction on total degradation of $\Delta \mathcal{L}$ is obeyed and ensures that only one configuration (of quantized weights and activation) per layer is selected.

### 3.3 BATCH NORMALIZATION TUNING

A common practice is fusing BN layers into their predecessor weight layers before applying post-training quantization to reduce the amount of Multiply-Accumulate (MAC) operations. However, the reduction in bit-width after quantization can cause the model's internal statistics to deviate further from those of the full precision model. To compensate for this deviation, we suggest updating BN statistics. First, we need to reconstruct the BN layers then re-tune the BN layers' statistics (by a few iterations of running-mean to re-collect the statistics). Finally, re-absorb (re-fuse) the BN layers into the weight layers (this is possible only in a per-channel weights quantization setting, which is the current standard). Next, we give more details on each phase.

**Reconstructing BN layers** Assume the original (pre-fusing) BN parameters $\gamma_o, \beta_o$ and $\epsilon$ are known, as is usually the case. We would like to initialize $\mu, \sigma^2$, as well as the BN parameters $\gamma_r$ and $\beta_r$ ($r$ for "reconstructed") so that the reconstructed BN

$$BN_r(x) = \gamma_r \frac{x - \mu}{\sqrt{\sigma^2 + \epsilon}} + \beta_r \approx x \tag{6}$$

will re-adjust the model statistics. To do so, first we initialize the reconstructed BN layers by setting the following parameters (denoted by $r$):

$$\mu = \beta_r = \beta_o; \qquad \sigma^2 = \gamma_o^2 \qquad ; \gamma_r = \sqrt{\gamma_o^2 + \epsilon} \tag{7}$$

so that $BN_r(x) = x$. Then, we update $\mu$ and $\sigma^2$ by collecting running mean and running variance on the calibration data. We stress that the BN parameters, $\gamma_r, \beta_r$, do not change while applying BN tuning, as we only invoke forward propagation.

**Re-fusing BN layers** Due to the per-channel quantization setting we use, the collected statistics can be fused back into the current quantization scale as follows:

$$W_i' = W_i \frac{\gamma_r}{\sigma}; \qquad b_i' = \frac{\gamma_r}{\sigma}(b_i - \mu) + \beta_r; \qquad \Delta_{w_i}' = \frac{\gamma_r}{\sigma} \Delta_{w_i} \tag{8}$$

Thus, in addition to the regular BN fusion, the quantization step is adjusted by $\gamma_r \sigma^{-1}$. Additional details are given in section B of the Appendix .

**Bias tuning**    Much like Finkelstein et al. (2019), we suggest to apply a global bias-tuning procedure on the final mixed-precision model by applying quantization-aware training to minimize the Knowledge Distillation (KD) loss (which does not require labels). Since we restrict the trainable variables to be the biases only, we can train only on the calibration set without experiencing overfitting.

## 4    QUANTIZATION FLOW

Past years have seen the rapid development of efficient deployment techniques (Nagel et al., 2019; Haroush et al., 2019). Deployment flows can vary based on the user setting such as hardware constraints, deployment time and task/dataset availability. While some users are willing to pay at initialization the time and effort to gain another fraction of accuracy, others require a simple and fast solution. We address this by suggesting two novel pipelines, light and advanced. Our pipelines are designed to the current, most common setting: per-channel quantization with a small calibration set.

Our *light pipeline* requires three steps: (1) Fuse layers and define quantization parameters; (2) Find optimal mixed-precision configuration using IP; and (3) Use BN tuning to correct the internal statistics. We note that all steps do not require back-propagation and thus are very light and fast. In addition to the light setting, in the ***advanced pipeline*** we apply AdaQuant to reduce each layer's output distortion from its full precision counterpart before invoking the IP algorithm. A detail comparison between the two pipeline is given in table-1. Models that were optimized using AdaQuant to different bit-widths can be seamlessly stitched thus having the ability to create an optimized model in a mixed precision setting. Subsequently, global methods such as tuning both BN statistics and the layers' biases can be applied to reduce a Knowledge Distillation loss. Although there are additional post-training quantization techniques that could be potentially combined with our methods, such as bias correction (Banner et al., 2018), equalization (Meller et al., 2019), and outlier channel splitting (Zhao et al., 2019), we did not find it necessary: our results demonstrate that our relatively simple pipeline yields state of the art accuracy on both vision and text models, even without combining such methods. In the following sections we show our findings and give an ablation study that highlights the importance of each method and their combination.

|  | AdaQuant | Mixed-Precision (IP) | BN tuning | Bias Tuning |
|---|---|---|---|---|
| Light-Pipeline | ✗ | ✓ | ✓ | ✗ |
| Heavy-Pipeline | ✓ | ✓ | ✓ | ✓ |

Table 1: Comparison between light and advanced pipelines.

## 5    EXPERIMENTS

In this section, we demonstrate our methods and pipelines on several models and datasets. We first start by analyzing image recognition models such as ResNet18/50, MobileNet-V2, which were trained over the ImageNet dataset. Next, we demonstrate our method robustness by applying it on question answering task using the popular BERT model (Devlin et al., 2018), which was fine-tuned on SQuAD1.1 dataset (Rajpurkar et al., 2016). In all our experiments, we used a small calibration set taken from the training dataset. Unless stated otherwise, we applied asymmetric per-channel quantization (i.e. GEMLOWP Wu et al. (2016)) with quantized offset (i.e., zero point). Next, we analyze each method's strengths and weaknesses separately and argue for its validity. Additional implementation details can be found in section and the code are given in sections D E of the Appendix.

### 5.1    ADAQUANT

Recently several researchers suggested different types of MSE optimization. In most cases, the optimization was done per-tensor (i.e., for the weights and activations separately). Here we argue that by optimizing both quantization parameters and the weights jointly we can reduce the MSE even further and hence improve the accuracy as demonstrated in fig. 2b. In contrast to AdaRound (Nagel et al., 2020) which restricted the change of the weights to be within $\pm 1$ we allow the weights to change as needed. As can be seen in fig. 2a the weights indeed change their quantized value by more than one. Since our pipeline is focused on the mixed-precision setting we optimize each layer

separately to enable maximum flexibility when stitching the optimized models. Under that setting AdaQuant can be performed in parallel across all layers. However, since most recent papers do not show full compression-accuracy curves and only a few attempt 4-bit compression, we also compare our results to common fixed configurations using our *sequential-AdaQuant* flavor. While sequential AdaQuant cannot be parallelized or used for the mixed-precision setting it yields best-in-class results for all models tested as can be seen in table-2 and 3. For instance, on the extensively studied 8bit MobileNet-V2 (MobileNet-V2) topology we achieved 71.6% top-1 accuracy — less than 0.5% degradation compared to the full precision counterparts (71.9%).

| | RN-18 | RN-34 | RN-50 | RN-101 | RNext-50 | Inc-V3 |
|---|---|---|---|---|---|---|
| ACIQ* (Banner et al., 2018) | 64.5% | 69.1 | 68.1% | 68.1 | N/A | 60.4 |
| DFQ* (Nagel et al., 2019) | 57.1% | N/A | 64.5% | N/A | N/A | N/A |
| **AdaQuant** | 67.4% | 70.3% | 73.7% | 74.4% | 74.0 | 72.6% |
| **Sequential-AdaQuant** | **69.4%** | **71.7%** | **75.1%** | **75.5%** | **75.6%** | **73.4%** |
| FP32 | 71.97% | 73.3% | 77.2% | 77.3% | 79.22% | 77.4% |

Table 2: INT-4 quantization of weights and activations. Top-1 score over imagenet dataset for different post-training quantization methods. All layers were quantized to 4-bit except first and last layers which were set to 8-bit. Methods marked with (*) were implemented according to the paper. In all our experiments we apply per-channel quantization of the weights.

| | MobileNet-V2 (top-1) | BERT-Base-SQuad1.1 (F1) |
|---|---|---|
| min-max | 70.9% | 87.83% |
| DFQ (Nagel et al., 2019) | 71.2% | N/A |
| ZeroQ (Cai et al., 2020) | 72.91% | N/A |
| **AdaQuant** | **73.03%** | **88.35%** |
| **Sequential-AdaQuant** | **72.94%** | **88.45%** |
| FP32 | 73.03% | 88.81% |

Table 3: INT-8 quantization of weights and activations. A comparison with DFQ and naive quantization methods (which uses the channel's full dynamic range). In all our experiments we apply per-channel quantization of the weights and quantized all layers to 8-bit.

Testing the strength of this method on both vision and text topologies resulted in state-of-the-art results. As can be seen in table 3, on BERT-base model over SQuAD1.1 dataset (BERT-Base-SQuAD1.1) we managed to obtain 88.45% F1 score using just AdaQuant — less than 0.5% of its full precision counterpart (81.3%). Throughout our experiments, we avoided using any augmentation technique and follow the known validation set prepossessing. s

## 5.2 INTEGER PROGRAMMING

Our Integer programming formulation requires us to have two quantities per-layer: (1) loss degradation and; (2) performance improvement. Obtaining those quantities requires to invoke the model over a small calibration set $L$ times (once per layer) and measure the loss degradation and the performance gain. In our experiments, we set the performance value to be the number of parameters, but this measure could be changed to any additive measure. In all experiments, we used 1000 samples from the training set as our calibration set. Our setting considers only a mixture of 8-bit and 4-bit layers; to further test IP capabilities, we investigate a mixture of 2-4-8 bits as well. Unfortunately, since 2-bits quantization in post-training setting results in high degradation, the IP algorithm chose only mixture of 4-8 bits for compression ratio higher than 12.5%. Yet for 12.5% compression ratio, IP method found that by setting one layer to 2-bits while setting 8 smaller layers to 8-bits accuracy gains over 5.5% with respect to uniform 4-bit quantization. Also, by allowing a less hardware friendly setting where numerical precision can have the form of any integer between 2-8, yields the highest compression-accuracy ratio (fig. 3 - *relaxed advanced pipeline*).

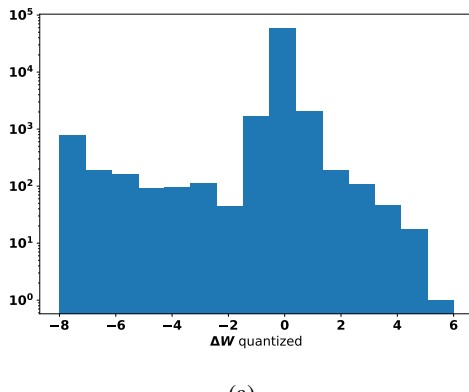 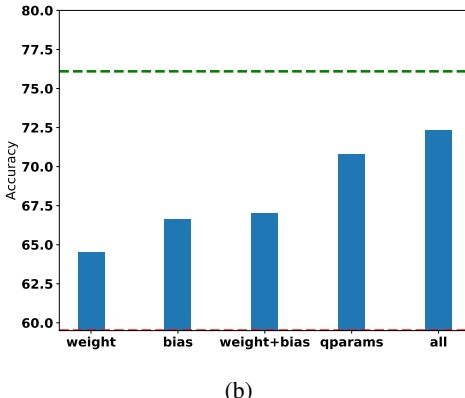

(a)                                          (b)

Figure 2: AdaQuant vs. AdaRound. (a) A histogram of $\Delta W$ distribution. AdaRound restricts this additive term to be $\Delta W = \pm 1$. Relaxing this constraint provides a more powerful optimization. (b) Ablation study on parameters optimization for ResNet50 over ImageNet. AdaRound is based exclusively on weight optimization, while AdaQuant optimizes the weights, biases, and other quantization parameters jointly.

### 5.3 BATCH-NORM TUNING

Batch-Norm Tuning (BNT) has a significant advantage, as it does not require weight optimization. Since BNT is applied by invoking the entire model, we must apply it only after setting the mixed-precision bit-width configuration. This is the case for all global optimization methods including bias-tuning. Notably, BNT requires only a few (at most 10) forward passes over the calibration set and yield significant gains (fig. 3). In this study, we applied BNT on models trained with BN layers only. However, it might be possible to extend this method to models without BN layers by reconstructing it from the statistics. We encourage the reader to investigate this path.

### 5.4 FULL PIPELINE AND ABLATION STUDY

Although several researchers suggested different methods for post-training mixed-precision quantization, none offer their code. Each paper focuses on a different quantization setting (e.g., quantizing only the weights, per-tensor quantization, etc.). Therefore, to demonstrate our pipeline strength, we created two different baselines based on common practices:

- Greedy-accuracy: recent studies suggested measuring each layer sensitivity and, based on the compression target, reduce the precision for the most robust layers.
- Greedy-compression: the complementary greedy approach (Lin et al., 2016) to sort the layers by their number of parameters and increase the precision of the layers from the smallest to the largest layer until the compression budget is reached.

Surprisingly, although the size of the layer should correlate with its sensitivity to quantization, the two greedy methods yield entirely different configurations. Investigating the configuration *greedy-compression* found that sorting by compression correlates with the location of the layers in the model. In most vision models, the layers closer to the input have fewer parameters. This aligns with current common practice (Banner et al., 2018). Notably, even when not combined with any other technique, the IP method obtained the best bit-width configurations stressing its importance.

Next, we turn to consider the light and advanced pipelines. Under challenging compression rates, our light-pipeline results highlight the importance of BN tuning. As can be seen in our experiment fig. 3, by merely invoking the model at inference mode for a few iterations and fixing the intermediate statistics, one can recover more than 1.5% of the accuracy (73.7% v.s 75.37%). As expected, by applying the advanced pipeline, one can obtain state-of-the-art accuracy. Arguably, our most impressive results are at 0.13% compression rate in which we managed to stay within 1% of the full precision accuracy while converting 96% of the model to 4-bit. For the challenging MobileNet-V2

we managed to switch 25% of the layers to 4bit (weights and activations) while maintaining less than 2% degradation; Additionally, we achieved, for the first time, reasonable top-1 accuracy of 65% when almost the entire model is in 4-bit.

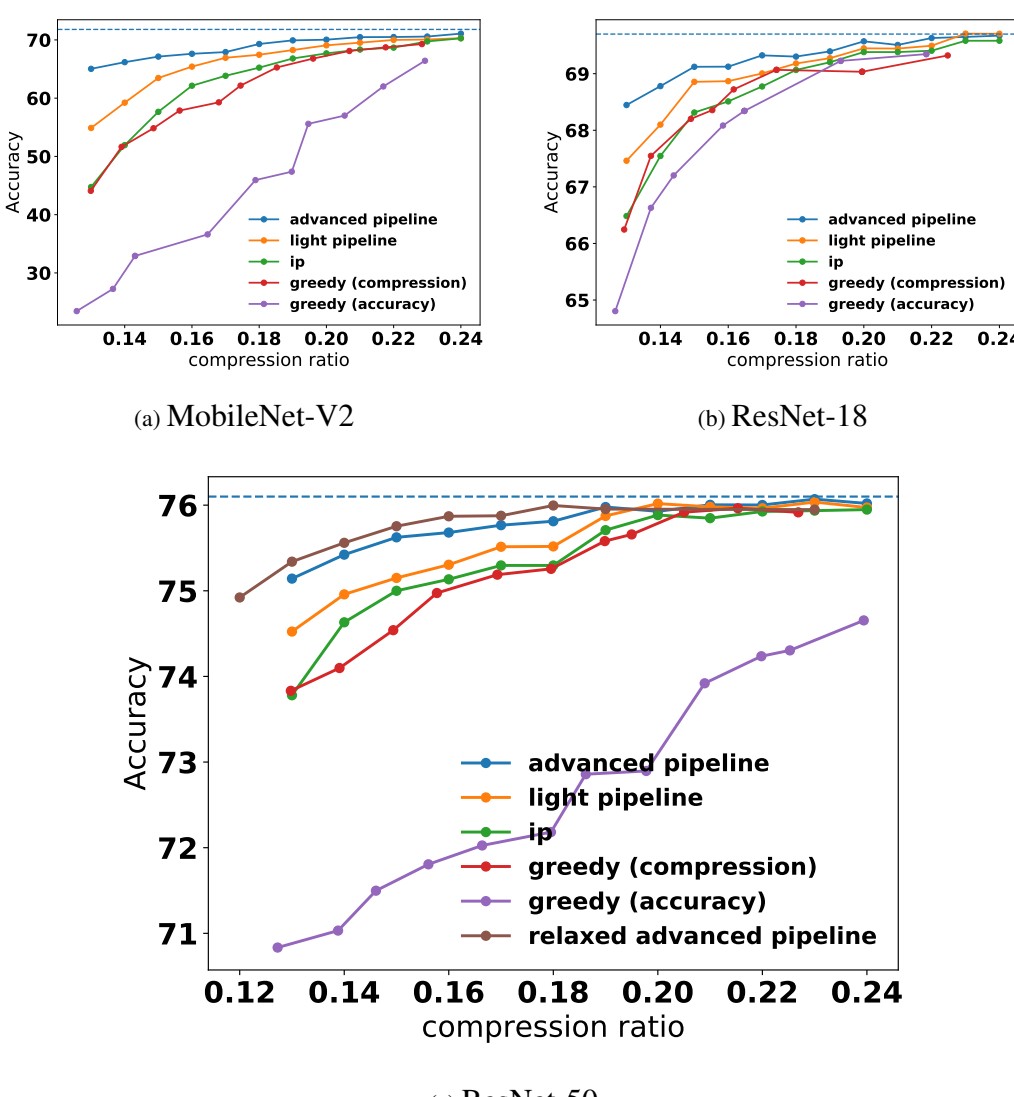

(a) MobileNet-V2          (b) ResNet-18

(c) ResNet-50

Figure 3: Ablation study over ResNet-50/18 and MobileNet-V2 - compression-accuracy curves. Our **advanced pipeline** is consist of AdaQuant, IP-mixed-precision, BN-tuning and bias-tuning. Our **light pipeline** is consist of only IP-mixed-precision, BN-tuning. The **relaxed advanced pipeline** appears in c is similar to the advance pipeline but allows the integer-programming to choose any bit-width between 2-8 and not just 4-bit or 8-bit. The compression ratio is measured as the ratio between the compressed model and the full-precision (32-bit) mode thus 0.25 compression rate indicate that the entire model uses 8-bit precision and respectively for 4-bit the compression rate is 0.125

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

# Appendix

## A  SIZE OF CALIBRATION SET

**Fully Connected layers**   Let's assume that we have weights of size $W \in R^{M \times N}$ and input and output are of sizes $N$ and $M$ respectively. Recalling Eq. 2 and setting $Y = WX$ and $W' = W + V$ results in:

$$\left(\hat{\Delta}_{w'}, \hat{\Delta}_x, \hat{V}\right) = \underset{\Delta_w, \Delta_x, V}{\arg\min} ||Y - Q_{\Delta_{w'}}(W') \cdot Q_{\Delta_x}(X)||^2,$$

For simplicity we assume that $\Delta_x$ is fixed and define $X_q = Q_{\Delta_x}(X)$, $W_q = Q_{\Delta_{w'}}(W')$. Therefore, if we have $B$ unique samples, then the problem we aim to solve have the following structure:

$$\begin{pmatrix} w_{11} & ... & w_{1N} \\ ... & \ddots & ... \\ w_{M1} & ... & w_{MN} \end{pmatrix} \begin{pmatrix} x_{11} & ... & x_{1B} \\ ... & \ddots & ... \\ x_{N1} & ... & x_{NB} \end{pmatrix} = \begin{pmatrix} y_{11} & ... & y_{1B} \\ ... & \ddots & ... \\ y_{M1} & ... & y_{MB} \end{pmatrix}$$

which translates to:

$$\begin{pmatrix} w_{11}x_{11} & ... & w_{1N}x_{N1} \\ w_{11}x_{11} & ... & w_{1N}x_{N2} \\ ... & \ddots & ... \\ w_{M1}x_{1B} & ... & w_{MN}x_{NB} \end{pmatrix} = \begin{pmatrix} y_{11} \\ y_{12} \\ \vdots \\ y_{MB} \end{pmatrix}$$

Notice that in the above equations, for each output we have a different set of parameters, therefore we can examine each output separately. For a single output we are in scalar linear regression with $N$ parameters and $B$ equations. If $B \geq N$ we are under-parameterized, and if $B < N$ we are over-parameterized.

**Convolution layers layers**   Similarly, for convolution layers with $C_o$ output channels, $C_i$ input channels, and kernel size $k$ each element of the output is a dot product of $C_i \cdot k \cdot k$ parameters. We have in total $Co \times H \times W$ outputs where $H, W$ is the output height and width. Thus we need $B \geq \frac{C_i \cdot k^2}{HW}$ samples to avoid over-fitting, where $B$ is the number of unique samples.

## B  RECONSTRUCTION AND RE-FUSING OF BATCH NORMALIZATION

In this section, we provide more details on Batch Normalization reconstruction and re-fusing procedure.

**Reconstructing BN layers:** Consider a Batch Normalization layer with parameters $\gamma_o, \beta_o$ that fused into previous convolutional layer weight and bias. Fusing batch normalization layer transforms weights and bias as following:

$$W'_i = W_i \frac{\gamma_o}{\sigma}; \qquad b'_i = \frac{\gamma_o}{\sigma}(b_i - \mu) + \beta_o; \tag{9}$$

To reconstruct the batch normalization, we would like to initialize $\mu, \sigma^2$, as well as the BN parameters $\gamma_r$ and $\beta_r$ ($r$ for "reconstructed") so that the reconstructed BN is approximately identity fig. 4.

$$BN_r(x) = \gamma_r \frac{x - \mu}{\sqrt{\sigma^2 + \epsilon}} + \beta_r \approx x \tag{10}$$

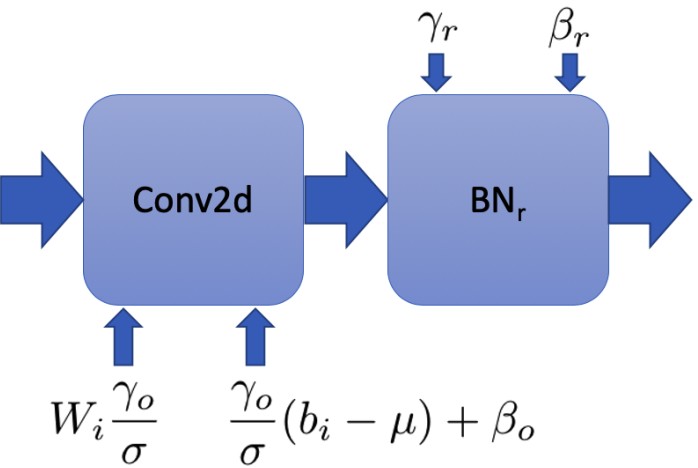

Figure 4

To do so, first we initialize the reconstructed BN layers by setting the following parameters (denoted by $r$):

$$\mu = \beta_r = \beta_o; \qquad \sigma^2 = \gamma_o^2 \qquad \gamma_r = \sqrt{\gamma_o^2 + \epsilon} \tag{11}$$

so that $BN_r(x) = x$.

Now, we can update $\mu$ and $\sigma^2$ by collecting running mean and running variance on the calibration data. We stress that the BN parameters, $\gamma_r, \beta_r$, do not change while applying BN tuning, as we only invoke forward propagation.

**Re-fusing BN layers:** After BNT phase we need to fuse Batch Normalization layer again into convolution weights and bias. Regular batch normalization fusing will cause degradation due to quantization of the weights. To resolve this issue we can leverage per-channel quantization setting we use.

Denote $s_{w_i}, z_{w_i}$ scale and zero point of the weigh, the quant/dequant operation defined as:

$$W_q = s_{w_i} \left( \left\lfloor \frac{W}{s_{w_i}} - \left\lfloor \frac{z_{w_i}}{s_{w_i}} \right\rceil \right\rceil + \left\lfloor \frac{z_{w_i}}{s_{w_i}} \right\rceil \right) \tag{12}$$

We can fuse parameters of the batch normalization layer as following:

$$
\begin{aligned}
W'_i &= W_i \frac{\gamma_r}{\sigma_x}; \qquad b'_i = \frac{\gamma_r}{\sigma_r}(b_i - \mu_x) + \beta_r \\
s'_{w_i} &= \frac{\gamma_r}{\sigma_x} s_{w_i}; \qquad z'_{w_i} = \frac{\gamma_r}{\sigma_x} z_{w_i}
\end{aligned}
\tag{13}
$$

Finally we can show that transformations eq. (13) equivalent to $\frac{\gamma_r}{\sigma_r} W_q$

$$
\begin{aligned}
W_q' &= s_{w_i}' \left( \left\lfloor \left| \frac{W'}{s_{w_i}'} - \left\lfloor \frac{z_{w_i}'}{s_{w_i}'} \right\rfloor \right| \right\rfloor + \left\lfloor \frac{z_{w_i}'}{s_{w_i}'} \right\rfloor \right) \\
&= \frac{\gamma_r}{\sigma_r} s_{w_i} \left( \left\lfloor \left| \frac{W}{s_{w_i}} - \left\lfloor \frac{z_{w_i}}{s_{w_i}} \right\rfloor \right| \right\rfloor + \left\lfloor \frac{z_{w_i}}{s_{w_i}} \right\rfloor \right) \\
&= \frac{\gamma_r}{\sigma_r} W_q
\end{aligned}
\tag{14}
$$

## C  ADDITIVE LOSS ASSUMPTION FOR INTEGER-PROGRAMMING

Suppose the loss function of the network $\mathcal{L}$ depends on a certain set of variables (weights, activations, etc.), which we denote by a vector $\mathbf{v}$. We would like to measure the effect of adding quantization noise to this set of vectors.

Since the quantization is emulated with additive noise, the loss is smooth and thus can be expanded to the Taylor series:

$$
\Delta \mathcal{L} = \mathcal{L}(\mathbf{v} + \boldsymbol{\varepsilon}) - \mathcal{L}(\mathbf{v}) = \tag{15}
$$

$$
= \frac{\partial \mathcal{L}^{\mathcal{T}}}{\partial \mathbf{v}} \varepsilon + \varepsilon^T \frac{\partial^2 \mathcal{L}}{\partial^2 \mathbf{v}} \varepsilon + O\left( \|\boldsymbol{\varepsilon}\|^3 \right). \tag{16}
$$

One can see from Eq 16 that when the quantization error $\varepsilon$ is sufficiently small, the overall degradation $\Delta \mathcal{L}$ can be approximated as a sum of $N$ independent degradation processes by neglecting the quadratic terms $\varepsilon^2$:

$$
\Delta \mathcal{L} \approx \frac{\partial \mathcal{L}^{\mathcal{T}}}{\partial \mathbf{v}} \varepsilon = \sum_i^n \frac{\partial \mathcal{L}}{\partial v_i} \cdot \varepsilon_i \tag{17}
$$

We note that Lin et al. (2016); Choukroun et al. (2019) used a similar assumption with respect to the additivity of quantization noise.

## D  EXPERIMENTAL DETAILS

In all our experiments, we used a small subset of the training set to run our methods. Specifically, for vision models, we used 1000 unlabeled images from the ImageNet training set (single image for each class) as a calibration set. For the Bert model, we used one paragraph from the training set. All presented methods AdaQuant, BNT, BT, and IP, performed well on such small calibration set producing SOTA results. Next we detail our setting for each of the technique in our pipelines

### D.1  ADAQUANT

AdaQuant optimization problem defined as following except zero-point of the quantizer which we omitted from eq. (18):

$$
\left( \hat{\Delta}_w, \hat{\Delta}_x, \hat{V}_W, \hat{V}_b \right) = \underset{\Delta_w, \Delta_x, V_W, V_b}{\arg\min} \| WX + b - Q_{\Delta_w}(W + V_W) \cdot Q_{\Delta_x}(X) - Q(b + V_b) \|^2 \tag{18}
$$

Technically to find a solution for eq. (18), we use Adam optimizer with different learning rates per type of parameters. We set different learning rates for weight, bias, and quantization parameters of input and weights. After experimenting with different models, we found that the same set of LR parameters worked for each model. The learning rates are $1e-5, 1e-3, 1e-1, 1e-3$ for weight, bias, quantization parameters of the inputs, and weights, respectively.

For vision models, we used 1000 unlabeled images from the ImageNet training set (single image for each class), running Adam optimizer for 100 iterations and a batch-size of 50 unless otherwise stated.

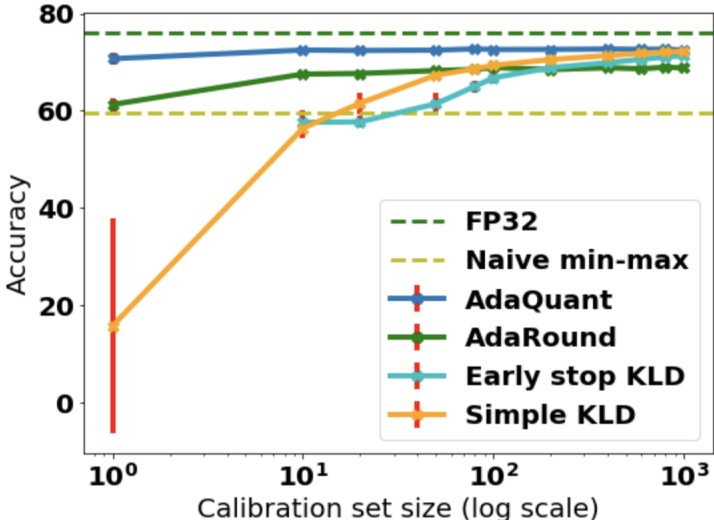

Figure 5: Calibration size ablation study with additional early-stop plot.

For BERT-base model, we used one paragraph from the training set, running Adam optimizer for 50 - 100 iterations depending on the type of layer. Learning rates and batch size are the same as of vision models.

In fig. 1 we aimed to answer the following question: *Assuming you have a small calibration set and no resources constraints (time,power) which method is the most accurate and robust* Out method were evaluated by running each experiments five times and reporting mean and standrad deviation. Here, in fig. 5, we add an additional naive early-stop plot on top of QAT-KLD experiment. We split the calibration data into two equal sets and train on half the examples while evaluation our performance on the other half Both *KLD* experiments used an SGD optimizer over 10 epochs; starting with learning rate of 0.1 and decreasing it by 1e-2 factor after 2 and 8 epochs. We also conducted *KLD* experiments with Adam optimizer and learning rate of 1e-3 where performed but their results were inferior. As can be seen in the plot AdaQuant is superior to other methods and remarkably excels on small calibration sets. As can be seen in fig. 5 the early exit results were inferior to the QAT-KLD as they use much smaller training set. However, other types of training-validation splits (e.g. 80-20) may boost the results.

### D.2 INTEGER PROGRAMMING

Our IP method requires two steps, the first is measuring the properties of each layer, and the second is applying the program based on these measurements with user defined constraint. As reference, we measure the loss (can also be accuracy) of the base precision model on the calibration set. Next, we measure the sensitivity of each layer by evaluating a model where all layers are qunatize to the base-precision but one layer that is quantized to lower precision (e.g., all 8-bit but one layer with 4-bit). The $\Delta \mathcal{L}_l$ in Eq. 3 is defined as the difference between the reference model loss and the measured loss. If a layer is robust to quantization, $\Delta \mathcal{L}_l$ will be small, and if a layer is sensitive to quantization, $\Delta \mathcal{L}_l$ will be large. The performance gain in the case of compression, is simply the model parameters size difference when lowering the precision of the examined layer. Hence, if a layer has $N$ parameters, the performance gain when changing from 8-bit to 4-bit result in compression gain of $\Delta \mathbb{P}_l = N * 8 - N * 4 = 4N$. In the second stage, we run the integer program based on the sensitivity and compression measured on each layer along with the user defined constraint.

### D.3 BATCH NORMALIZATION AND BIAS TUNING

The Batch Norm tuning phase is the most lightweight phase of the pipeline. We found empirically less than ten iterations of statistics update are sufficient. We also found that as compression growth,

more iterations of batch norm tuning are required. At the bias tuning phase, we perform 200 iterations of fine-tuning with the learning-rate of $0.1$.

## E    CODE

For all our vision dataset we used the default *torchvision* pre-trained model. For BERT-base experiment we fined-tuned on SQUAD1.1 dataset and provide the script for that as a part of our repository. Our code can be found at: *https://github.com/papers-submission/CalibTIP*.

