# OpenReview forum: "Improving Post Training Neural Quantization: Layer-wise Calibration and Integer Programming"
_ICLR.cc/2021/Conference — Reject_

### Official Review · AnonReviewer1 · 2020-10-27
**Interesting Approach To Quantization**

**Rating:** 7
**Confidence:** 5

**Review:**



Summary: The paper studies the problem of Post-Training Quantization of NNs, where no fine-tuning is performed to quantize the model. In particular, the authors focus on sub-8 bit quantization and propose a novel integer linear programming formulation to find the optimal bit width for a given model size. Additional approaches are proposed to minimize accuracy degradation after quantization. These include
(i) AdaQuant in which the parameters are quantized layer-by-layer to match the full precision output,
(ii) a batch norm tuning approach to re-adjust the statistics to the quantized model, and
(iii) an advanced pipeline for cases where backpropogation can be performed.

Experiments are performed on ResNet18/50, MobileNetV2, and BERT-SQuaD1.1 showing the feasibility of the proposed method.

I think the approach in the paper is pretty interesting, and specially the integer linear programming solution.
Overall the paper is strong however, please note the following:


- Page 3 last paragraph: it seems there are errors in the results for calibration data.
1/ it is stated that if "B>> M then we might underfit the data.". Do the authors mean "B>>N"? Similarly it seems the result given for the convolution (i.e. B> cK^2/HW) needs to be revised.
2/ The analysis requires specifying the rank of W and in particular the relationship between M and N. In particular, note that the matrix W can at most have a rank of min(M,N) and as a result the rest of the linear equations for calibrating the data would be redundant. This needs to be taken into account in your result.

- Page 4: "Depending on the needs, our performance metrics P would be either the execution time of the network or its power consumption." This is good but no result on either latency or power consumption is provided in the paper.

- Figure 4 (a,b): Please provide the BOPS for the mixed-precision results. BOPS proposed by (https://arxiv.org/pdf/2005.07093.pdf) is a good metric to measure the total reduction in computations for mixed precision quantization.

- The results section uses weak FP32 baseline. For instance, the baseline accuracy for ResNet50 is 77.2\% and MobileNetV2 is 73\%, while the paper uses 76.1\% and 71.8\%.

- Related to the above, it is stated "on the extensively studies 8bit MobileNet-V2 topology we achieved 71.6% top-1 accuracy". This is a good result but please note that other work in the literature (arxiv:2001.00281) reports 72.91\% for INT8 quantization of MobileNetV2 (this comparison is actually missing from the paper). It is immediately not clear if the lower reported accuracy is due to the weaker FP32 baseline used or if it is an inherent problem with the method (most probably it is the former but it would be to show this).




Minor Comments:

- There were several grammatical/spelling mistakes in the paper. Please proofread the paper thoroughly. Below are some of the errors that I caught:

- 3 OPTIMIZING QUANTIZATION PIPLINE -> PIPELINE

- Page 6:  the our method robustness -> the robustness of our method

- Page 6: manged -> managed

- Page 7: an significant advantage -> a significant advantage

- Page 8: we managed to switched  -> we managed to switch

- Page 8: For instance, on the extensively studies -> For instance, on the extensively studied

---

> ### Author Response · Authors · 2020-11-21
> **Reply to AnonReviewer1**
>
> We thank the reviewer for his positive and encouraging feedback. Please refer to our answers below
>
> Q: Page 3 last paragraph: it seems there are errors in the results for calibration data. (1) it is stated that if "B>> M then we might underfit the data.". Do the authors mean "B>>N"? Similarly it seems the result given for the convolution (i.e. $B> cK^2/HW$) needs to be revised. (2) The analysis requires specifying the rank of W and in particular the relationship between M and N. In particular, note that the matrix W can at most have a rank of min(M,N) and as a result the rest of the linear equations for calibrating the data would be redundant. This needs to be taken into account in your result.
>
> A: (1) Thanks for catching this typo, corrected.
> (2) We respectfully disagree. First, let's consider for example the case of single output ($M=1$). Here we are in scalar linear regression with $N$ parameters and $B$ equations. So we are over-parameterized if and only if $B<N$. However, if we replace $N$ with $\min(N, M)$, as implied by the reviewer suggestion (if we understood it correctly), will results in $B< \min(M,N) = 1$,  thus we will never be over-parameterized, which is clearly incorrect. Next, when $M>1$, we can just look at each equation independently (i.e., each equation contains different variables), so the same conclusion remains. We added a more detailed discussion of this in Appendix A to clarify these issues.
>
> Q: Figure 4 (a,b): Please provide the BOPS for the mixed-precision results. BOPS proposed by (https://arxiv.org/pdf/2005.07093.pdf) is a good metric to measure the total reduction in computations for mixed-precision quantization.
>
> A: In this work we compared the accuracy to compression ratio. Changing the performance acceleration matrix to BOPS is relatively easy, but require to rerun all the experiments. We hope to add those to the appendix soon.
>
> Q: The results section uses a weak FP32 baseline. For instance, the baseline accuracy for ResNet50 is $77.2\%$ and MobileNetV2 is $73\%$, while the paper uses $76.1\%$ and $71.8\%$.
>
> A: We used the standard torchvision [1] models as given by pytorch but open your request we updated the models in section 5.1 to pytorchcv [2] models.
>
> Q: It is stated, "on the extensively studies 8bit MobileNet-V2 topology we achieved 71.6\% top-1 accuracy". This is a good result but please note that other work in the literature (arxiv:2001.00281) reports 72.91\% for INT8 quantization of MobileNetV2 (this comparison is actually missing from the paper). It is immediately not clear if the lower reported accuracy is due to the weaker FP32 baseline used or if it is an inherent problem with the method (most probably it is the former but it would be to show this).
>
> A: Thank you for pointing us to that result. We used Pytorch native models as given by torchvision, but upon your request, we update the results for pytorchcv model used by ZeroQ. AdaQuant results in full accuracy of 73.03\% top-1. We added the result to the paper in section 5.1 table-2.
> Minor Comments:
>
> Claim: There were several grammatical/spelling mistakes in the paper. Please proofread the paper thoroughly.
>
> A: Thanks for the feedback we fixed all the typos in the revised manuscript.
>
> [1] https://pytorch.org/docs/stable/torchvision/models.html
> [2] https://pypi.org/project/pytorchcv/

---

> > ### Comment · AnonReviewer1 · 2020-11-22
> > **Rebuttal Addresses My Concerns**
> >
> > I would like to thank the authors for the detailed rebuttal. The comments address my concerns. I think this is a very good paper.

---

### Official Review · AnonReviewer2 · 2020-10-27
**Collection of steps to deal with quantization-induced error in post-training quantization**

**Rating:** 6
**Confidence:** 3

**Review:**

This work presents a quite comprehensive multi-step scheme for post-training neural quantization that does not rely on large datasets or large computational resources.

The work is has significance in the domain of post-training neural quantization, especially in cases where only a small calibration set or limited resources are available. It would be interesting to also think about accumulator quantization.


Pros:

The empirical results are relatively strong in this method; 4-bit quantization is a good achievement in the models considered here.

The quantization process covers nicely the various different parts of the errors that  post-training quantization induces and propose somewhat original solutions to them.

Cons:

The framework is relatively complex and consists of multiple steps.

What is the detailed difference of computational resource use between light and advanced pipeline?

AdaQuant seems like a rather straight-forward step from AdaRound by combining with it some related works.

It is not clear how the BN error compensation differs exactly from related work.

Some details were missing, for example, it is up to the reader to guess how many bits were used in the accumulators.

Some spelling mistakes, e.g., “Optimizing Quantization Pipline”


Overall: An engineering oriented paper with some lack of testable hypotheses and analysis of some parts of the methods, but the 4-bit results justify publication. Edit: I have not seen author reply and further reading of the paper has not clarified the main issues found by all reviewers. I have to lower the score. Edit2: new version of the paper and the author reply cleared some concerns, score raised accordingly.

---

> ### Author Response · Authors · 2020-11-21
> **Reply to AnonReviewer2**
>
> We thank the reviewer for his positive feedback. Please refer to our answers below
>
> Claim: The framework is relatively complex and consists of multiple steps.
>
> A: The framework contains several components, and each can be applied separately. All components have almost no hyperparameters and require very little time to use. We believe that the code we created is easy to use and run and thus reproducing the results and extending it should be easy. For instance, upon request of Reviewer #5 we add experiments over different flavors of ResNet (R-34/R101) and ResNext101 as well as inception-v3.
>
> Q: What is the detailed difference of computational resource use between light and advanced pipeline?
>
> A: The advanced pipeline as opposed to the light pipeline starts with AdaQuant, which applies per-layer optimization for the weights and quantization parameters and ends with bias correction as detailed in table-1 in section 4. We measure the time it takes to run AdaQuant and it does not exceed 3m on CPU and 30sec on GPU. Since AdaQuant can be performed in parallel we believe this is a reasonable time. Bias correction requires approximately 10 epochs.
>
> Claim: AdaQuant seems like a rather straightforward step from AdaRound by combining with it some related works.
>
> A: While AdaRound performs a quite complex optimization to find the perfect rounding for each weight (by using soft and hard sigmoids), we use a much simpler and stronger approach to tune the weights. We used AdaQuant as an optimization step before mixed-precision. Since most works just use a fixed configuration (e.g., keep first and last layers in higher precision)  we added experiments with sequential AdaQuant (section 5.1 of the revised paper). Sequential AdaQuant fixed an inherent error of AdaRound as it enables correcting quantization errors obtain by quantizing predecessor layers. We kindly ask the reviewer to read this section - we believe the results are quite impressive.
>
> Claim: It is not clear how the BN error compensation differs exactly from related work.
>
> A: We agree that the fact that re-tuning BN-statistics improves performance is not surprising, however, it is not our novelty here. First, we describe how to reconstruct BN layers, which commonly do not exist in a post-training setting, due to folding. Without these reconstructed BN layers we could not have tuned the statistics. Second, we are the first to show that BN folding after BN tuning can be done in per-channel weight quantization (this is enabled by the per-channel weight scales). We believe that these findings and the detailed code would be appreciated by practitioners.
>
> Q: Some details were missing, for example, it is up to the reader to guess how many bits were used in the accumulators.
>
> A: We used 32bit accumulators but it is interesting to check 16bit as well.
>
> Claim: Some spelling mistakes, e.g., “Optimizing Quantization Pipeline”
>
> A: Thank you for your feedback we fixed all typos in the revised manuscript.

---

> > ### Author Response · Authors · 2020-11-23
> > **Author response  AnonReviewer2 - published Nov 21st**
> >
> > We noticed  #reviewer 2 recently edited the review: "I have not seen author reply and further reading of the paper has not clarified the main issues found by all reviewers."
> > Perhaps the reviewer missed we did reply to each reviewer in a separate reply (after we posted the general reply explaining the changes in the manuscript).
> > We kindly ask the reviewer to take a look at our response and the revised manuscript (published Nov 21st). We believe we have addressed all comments (if not, we would like to understand what is missing).

---

### Official Review · AnonReviewer4 · 2020-10-28
**See below**

**Rating:** 6
**Confidence:** 4

**Review:**

The paper introduces a series of techniques to quantize neural networks, and how to combine them:
* Layer by layer quantization where weights can change as needed (rather than to the nearest quantization error).
* Integer programming to determine the precision required at every layer.
* Tuning batch norm weights by re-computing statistics.

PROS

* Main text is well written (see below for other issues though).
* All components in the proposed method are straightforward.
* Strong results, with very little performance loss after very aggressive quantization (~4 bits).

CONS

* The organization of the paper can and should be easily improved (see below).
* Very basic details, such as the metric used in some tables, are omitted.
* Few experiments/baselines.

---

On the flow of the paper: Fig. 1 is barely introduced. Fig. 2 is introduced much earlier than it is first referred to in the text, in section 5.2, where it is not explained either. The labels in the figure are not defined either at this point (and ip should be IP). Additionally, the colors for the last two lines are too similar.

The experiment of Fig.1  should be introduced with some level of detail (there is none outside the caption). The caption is a bit confusing: what are the "the last layers after"? Why are variance bars always red? They are also hard to see. The labels are not descriptive (light pipeline, advanced pipeline, relaxed advanced pipeline) and hard to find in the text. MN-V2 and B-SQuad1.1 just drop in Table 1 without any context. It can be inferred from earlier sections what this refers to but it should be explained clearly. Same for "min-max" in Fig. 1 and Table 1.

Baselines are barely discussed. Also, I am not very familiar with quantization papers, so I might have missed relevant baselines, but they seem hard to compare. For instance, the authors omit [Nagel et al 2020](https://arxiv.org/pdf/2004.10568.pdf), which seems to do better at similar quantization levels, but I am not sure the results are directly comparable. The authors should discuss this better.

I think section 4 can be titled simply "Quantization flow".

Compression ratio in the plots should indicate %?

---

Typos/grammar:

* "Mobilsnet-V2" -> "Mobilenet-V2
* "we suggest [an] integer-linear programming"
* "AdaRound['s] implicit constraint"
* "MSE distance": shouldn't this just be MSE?
* "Early work by Lin et al. (2016) used [a] convex optimization formulation which results ~~with~~ [in] a simple greedy compression scheme."
* "3. OPTIMIZING [THE] QUANTIZATION PIP[E]LINE"
* "model['s] internal statistic[s]" -> found twice
* "often result with an inferior solution" -> "often result in an inferior solution"
* "Accordingly, researches suggested" -> researchers?
* "where V is a continuous variable V" -> redundant
* "thus enjoy[ing] some of the flexibility"? -> "benefitting from" might be a better phrasing?
* "Quantization-Aware- Training" -> extra space
* "and [is] much less prone to over-fitting"
* "[A] Similar derivation"
* "Even optimizing on [a] single image"
* "MAC operations." -> acronym not previously introduced (unless mistaken)
* Section 4: IP acronym should be introduced in the integer programming section.
* "we investigate [a] mixture"
* "results with high degradation" -> in high degradation

---

> ### Author Response · Authors · 2020-11-21
> **Reply to AnonReviewer4**
>
> We thank the reviewer for his positive and thorough feedback; please see our answers below:
>
> Q: Very basic details, such as the metric used in some tables, are omitted.
>
> A: Thank you for your feedback we fixed this in the revised manuscript.
>
> Claim: On the flow of the paper: Fig. 1 is barely introduced. Fig. 2 is introduced much earlier than it is first referred to in the text, in section 5.2, where it is not explained either. The labels in the figure are not defined either at this point.
>
> A: Thank you for your comment - we added more information about fig-1 in both section D of the appendix and manuscript. Figure-2 was moved to section 5 and additional details about the labels were added to the caption.
>
> Q: The experiment of Fig.1 should be introduced with some level of detail (there is none outside the caption). The caption is a bit confusing: what are the "the last layers after"?
>
> Why are variance bars always red? They are also hard to see. The labels are not descriptive (light pipeline, advanced pipeline, relaxed advanced pipeline) and hard to find in the text.
>
> A: Thank you for your feedback - we added details in the caption and fixed the "after" typo. The standard deviation was very small for calibration size larger than 100, thus we marked it in red.
>
> Claim: MN-V2 and B-SQuad1.1 just drop in Table 1 without any context. It can be inferred from earlier sections what this refers to but it should be explained clearly. Same for "min-max" in Fig. 1 and Table 1.
>
> A: Thank you for your feedback we fixed this in the revised manuscript.
>
> Claim: Baselines are barely discussed. Also, I am not very familiar with quantization papers, so I might have missed relevant baselines, but they seem hard to compare. For instance, the authors omit Nagel et al 2020, which seems to do better at similar quantization levels, but I am not sure the results are directly comparable. The authors should discuss this better.
>
> A: Thank you for your comment, Negal et al 2020 (AdaRound) only quantized the weights as opposed to AdaQuant which quantize both the weights and activations. For a fair comparison, we used our own implemented AdaRound based optimization and used it to quantize both weights and activation. As can be seen in Figure 1, Adaquant results with a significantly improved accuracy compared to AdaRound (Nagel et al 2020) for all considered calibration set sizes.
>
> A: Thanks for your detailed feedback- all typos where fixed in the manuscript and section 4 title was changed to "Quantization flow".

---

### Official Review · AnonReviewer3 · 2020-10-28
**Overall, the authors propose to use a combination of different twisted techniques for better neural quantization, which lacks meaningful insights.**

**Rating:** 4
**Confidence:** 2

**Review:**

The authors propose to use many techniques to push the limit of neural quantization, which shows reasonable improvements in some datasets.

+ good performance.
+ clear presentation.
+ easy to read and follow

My main complaint is that this type of combination is better for a technical report rather than a standalone paper. In more detail, the authors claim four proposed components: AdaQuant, Integer programming, Batch-norm tuning, and two pipelines for neural quantization. However, the problem is, are these four components all original or just adapted from existing works? The authors are failed to present the relation of the proposed components with the exitinging ones. For example, in my understanding, the AdaQuant is only an adapted version of AdaRound, but the discussion about it (in Section 3.1) is too superficial. Furthermore, the experiment part is not convincing and I think can not back up the claim. With so many "proposed components", the missing of thorough ablation study is a big problem.

---

> ### Author Response · Authors · 2020-11-21
> **Reply to AnonReviewer3**
>
> We thank the reviewer for his feedback, please see our answers below:
>
> Q: " However, the problem is, are these four components all original or just adapted from existing works?"
>
> A: All three components (AdaQuant, Integer programming, Batch-norm tuning) are original, and lead to significant improvements over previous works, as we discuss below.
>
> Claim: " The authors are failed to present the relation of the proposed components with the existing ones. "
>
> A: These relations are discussed in section 2 "Related works".
>
> Claim: "For example, in my understanding, the AdaQuant is only an adapted version of AdaRound, but the discussion about it (in Section 3.1) is too superficial. "
>
> A: Indeed, both AdaQuant and AdaRound minimize the per-layer reconstruction error. However, AdaQuant is much less restrictive, allowing the weights to have large changes, while AdaRound’s forces the quantized weights to be within 1 of their round-to-nearest value.  In addition, AdaRound only quantized the weights and not the activation's thus it is not optimized the quantization parameters.
>
> Claim: "Furthermore, the experiment part is not convincing and I think can not back up the claim. With so many "proposed components", the missing of thorough ablation study is a big problem."
>
> A: Please note figure 1, where we show AdaQuant (without any other method) leads to major improvements in test accuracy over AdaRound --- even when optimized over very small calibration data sets. We find this to be quite remarkable and unusual since fine-tuning the entire model over a small calibration set is known to overfit the data. Additionally, in figure 3 we show how the addition of Integer programming and BN tuning each improves performance. Finally, we suggest sequential AdaQuant as described in section 5.1 and suggest light and advanced pipelines for the mixed-precision setting.

---

### Official Review · AnonReviewer5 · 2020-11-09
**Confusing claims with lack of details and limited novelty**

**Rating:** 4
**Confidence:** 5

**Review:**


This paper proposed a set of methods for post-training quantization of dnns. The methods include AdaQuant (which jointly optimizes quantization steps for weight and activation per output activation of each layer), Integer Programming (which determines bit-precision for all the layers), and the batchnorm tuning. The authors presented promising experimental results on various neural networks to support the proposed methods.

However, there are serious concerns about these claims as follows:
1) AdaQuant
- It is straightforward to think that the joint optimization of quantization step size for weight and activation would result in better quantization results. But this joint optimization would also increase the search space (at least) quadratically, resulting in significant computational cost. Note that the biggest merit of post-training quantization is its simplicity (cf., QAT incurs full-blown training epochs); thus increased cost for post-training quantization is not desirable. Since AdaQuant is the major claim, the authors should provide more discussion on how they dealt with this increased complexity.

- The authors claim that AdaQuant avoids overfitting, but the reason does not seem to be clear. There is no clear explanation of how AdaQuant increases the generality of the quantized model, and the discussion about the sample size (B) is hard to understand (why there's infinite solution when B << N? how B>= Ck^2/(HW) is derived for the convolution case?)

- Also, it seems that the "per-channel" quantization method is utilized in this work, but the formulation in (2) seems to be for "per-layer" optimization. Why they are different?

- How much time does it take to solve this joint optimization?


2) Integer Programming
- The authors proposed an Integer Programming formulation, but there seem to be missing information: what is the formulation of the penalty function, "deltaL"? The authors described it simply as "Loss", but it is not clear what the exact method it is calculated. In fact, deltaL can be pretty complex functions, which might not be independent terms for each layer; thus the formulation like (3) might not be correct. Note that the impact of quantization in the earlier layers affect the quantization impact in the current layer. Without clear explanation and justification about it, the proposed IP formulation does not make sense.

- The authors mentioned that deltaP should be additive and sum up to the total benefit. How can one guarantee it?

- Also, it seems that the complexity of the IP optimization increases as the number of layers increases. How much computation time increases if the number of layers are large?


3) Batch normalization tuning
- Unfortunately, there is a very similar idea proposed by [Sun et al., NeurIPS 19]. Cf. "Sec.3 Trans-Precision Inference in FP8".


Also, there are several suggestions to improve understanding of readers.
- Currently, the ablation study looks very confusing. It is not clear which of the pipeline options (light, advanced?) include what kinds of techniques. Please do specify (maybe in a separate table) the list of techniques covered by different pipeline options.

- The proposed method is not much evaluated by various neural networks. It would be desirable to expand the coverage of neural nets as much as the prior work did.

- Currently, the proposed methods only utilized "per-channel" quantization. How much accuracy the proposed methods can maintain if they adopt "per-layer" quantization?

- What is the definition of "compression ratio"? (typically compration RATIO is like 12:1, and compression rate is like 2X, 3X...)

-

---

> ### Author Response · Authors · 2020-11-21
> **Reply to AnonReviewer5 - Part 1**
>
> We thank the reviewer for his feedback. Please see our answers below:
>
> Q: It is straightforward to think that the joint optimization of quantization step size for weight and activation would result in better quantization results. But this joint optimization would also increase the search space (at least) quadratically, resulting in significant computational cost. Note that the biggest merit of post-training quantization is its simplicity (cf., QAT incurs full-blown training epochs); thus increased cost for post-training quantization is not desirable. Since AdaQuant is the major claim, the authors should provide more discussion on how they dealt with this increased complexity
>
> A: We note that Eq.2 was optimized with Adam, a variant of SGD, as stated in section D of the appendix.  Such gradient-based optimization methods are known to work well even with very large parameter spaces (e.g. deep neural networks).  Thus, we are sure what is the concern regarding increased search space. We wish to stress that AdaQuant is extremely fast: in less than two minutes (using GTX 1080Ti) we obtain an optimized ResNet50 model. Over CPU it takes less than an hour. Furthermore, our current code can be easily improved by executing Adaquant on all layers in parallel. Per-layer Adaquant takes less than three minutes on CPU and 5 sec on GTX 1020Ti. Results were obtained with a calibration size of 100 samples. Additionally, we ran experiments with sequential AdaQuant which accounts for the quantization error from previous quantized layers and thereby improves results drastically. For instance, for ResNet50 we have been able to achieve 75.08\% top-1 accuracy when the entire model is in int4 (except the first and last layers, which were kept in 8-bit) as opposed to 73.7\% when running AdaQuant in parallel.
>
> Q: There is no clear explanation of how AdaQuant increases the generality of the quantized model, and the discussion about the sample size (B) is hard to understand (why there's infinite solution when $B << N$? how $B>= C_i k^2/(HW)$ is derived for the convolution case?)
>
> A: We added a detailed explanation in both the manuscript (section 3.1 ) and in section A of the appendix. For a Fully Connected layer with an input size of $B\times N$ and a weight matrix of size $M\times B$, the output size would be $B\times M$. Since we try to minimize the MSE over all the outputs we essentially have $BM$ equations with $NM$ parameters. In a generic linear system, if we have more parameters than equations, then we have an infinite number of solutions. However, some of those degrees of freedom can lead to over-fitting the data. Similarly, if we have many more equations than parameters we might result in under-fitting the data. To avoid this we wish to have roughly the same number of equations and parameters. A toy example can be found in section A of the Appendix. Similar derivation for convolution results with $BC_oHW$ equations and the total number of parameters is $C_oC_ik^2$.
>
> Q: It seems that the "per-channel" quantization method is utilized in this work, but the formulation in (2) seems to be for "per-layer" optimization. Why they are different?
>
> A: By per-layer optimization, we mean that each layer is optimized separately. Thus we do not have to run backpropagation on the entire model. As stated above this also implies that we can run AdaQuant in parallel on all layers. The paper applies per-channel quantization on the weights (i.e., a different scale for each kernel) and per-tensor quantization on the activation (i.e., one scale for all the activation tensors), as is commonly supported by hardware [1].
>
> Q: What is the formulation of the penalty function, "deltaL"? The authors described it simply as "Loss", but it is not clear what the exact method it is calculated. In fact, deltaL can be pretty complex functions, which might not be independent terms for each layer; thus the formulation like (3) might not be correct. Note that the impact of quantization in the earlier layers affect the quantization impact in the current layer. Without a clear explanation and justification about it, the proposed IP formulation does not make sense.
>
> A: You are right, for IP to work we had to assume that both the performance degradation and the loss degradation are additive. Clearly, for the loss, this is only a first-order approximation of its Taylor expansion. Similar to previous works that examined this approximation [2,3], we found it to be quite accurate for most models.
>
> [1] Jain, Animesh, et al. "Efficient execution of quantized deep learning models: A compiler approach." arXiv preprint arXiv:2006.10226 (2020).
>
> [2] Choukroun, Yoni, et al. "Low-bit quantization of neural networks for efficient inference." 2019 IEEE/CVF International Conference on Computer Vision Workshop (ICCVW). IEEE, 2019.
>
> [3] Lin, Darryl, et al., "Fixed point quantization of deep convolutional networks." International conference on machine learning. 2016.

---

> > ### Author Response · Authors · 2020-11-21
> > **Reply to AnonReviewer5 - Part 2**
> >
> > Q: The authors mentioned that deltaP should be additive and sum up to the total benefit. How can one guarantee it?
> >
> > A: The Performance gain, deltaP, is slightly more tricky to evaluate, as it might depend on the hardware and software at hand (e.g., latency and throughput). Yet for compression (reducing model footprint) it is obviously additive.  We added an explanation about this assumption in section C of the Appendix.
> >
> > Q: It seems that the complexity of the IP optimization increases as the number of layers increases. How much computation time increases if the number of layers is large?
> >
> > A: The computation increase linearly with the number of layers. Yet, evaluating the performance gain and loss degradation can be done in parallel for all layers. Additionally, our integer-programming exploits the well-optimized pulp library [4] thus requires only a few seconds to derive an optimal solution.
> >
> > Claim: Batch normalization tuning: unfortunately, there is a very similar idea proposed by [Sun et al., NeurIPS 19]. Cf. "Sec.3 Trans-Precision Inference in FP8".
> >
> > A: Thank you for referring us to this paper. We agree that the fact that re-tuning BN-statistics improves performance is not surprising, however, it is not our novelty here. First, we describe how to reconstruct BN layers, which commonly do not exist in a post-training setting, due to folding. Without these reconstructed BN layers we could not have tuned the statistics. Second, we are the first to show that BN folding after BN tuning can be done in per-channel weight quantization (this is enabled by the per-channel weight scales). We believe that these findings and the detailed code would be appreciated by practitioners.
> >
> > Q: Currently, the ablation study looks very confusing. It is not clear which of the pipeline options (light, advanced?) include what kinds of techniques. Please do specify (maybe in a separate table) the list of techniques covered by different pipeline options.
> >
> > A: In section 4 we detail the exact stages of each of the pipelines: "Our \textbf{light pipeline} requires three steps: (1) Fuse layers and define quantization parameters; (2)Find optimal mixed-precision configuration using IP; and (3) Use BN tuning to correct the internal statistics. ... in the \textbf{advanced pipeline} we apply AdaQuant to reduce each layer’s output distortion from its full precision counterpart before invoking the IP algorithm". We also added a table in section 4 to further explain the differences.
> >
> > Claim: The proposed method is not much evaluated by various neural networks. It would be desirable to expand the coverage of neural nets as much as the prior work did.
> >
> > A: Thank you for your feedback we added additional models (ResNet 34/101, ResNext50, and Inception V-3) in section 5.1.
> >
> > Q: Currently, the proposed methods only utilized "per-channel" quantization. How much accuracy the proposed methods can maintain if they adopt "per-layer" quantization?
> >
> > A: In this work we have a per-channel scale for the weights, while the activations have only one scale for the entire tensor (i.e. layer), as is commonly done [1]. However, our method can help per-tensor quantization as well. For instance on BERT-large with per-tensor quantization, we achieved 80.4\% exact-match vs. 79.6\% without traditional PTQ methods.  Our code supports per-tensor quantization (just change perC to False). Finally, please note that BN-tuning as a second stage after mixed-precision is not applicable for per-tensor quantization.
> >
> > Q: What is the definition of "compression ratio"? (typically compression RATIO is like 12:1, and compression rate is like 2X, 3X...)
> >
> > A: Compression ratio is relative to 32bit. Thus for 8bit (marked as 0.25), it is 1:4 and for 4-bit (marked as 0.125) it is 1:8. The rest is in between. We clarify this in the revised paper.
> >
> > [4] https://github.com/coin-or/pulp

---

### Author Response · Authors · 2020-11-21
**General Response**

We thank the reviewers for their positive feedback and insightful suggestions. The feedback helped us improve the manuscript and boost the suggested method performance.
We updated the manuscript as follows:
1. In section 3.1 we added a new flavor of AdaQuant, named sequential-AdaQunat, targeted for the common fixed bit allocation configuration (i.e., not mixed-precision setting)
2. In section 4.1 we added a table that clarifies the difference between the advanced and light pipelines.
2. In section 5.1, we updated baselines and added new results of sequential-AdaQuant - we believe they are quite impressive.
3. Several clarifications to the appendix including (a) the required size of calibration set; and (b) the additive assumption at the core of our integer-programming formulation.

We updated the code to include sequential-AdaQunat

We kindly ask the reviewers to consider our clarifications and improvements to the manuscript in their final score.

---

### Decision · Program_Chairs · 2021-01-07
**Final Decision**

**Decision:**

Reject

**Comment:**

This paper received mixed reviews, 3 positives (7, 6, 6) and 2 negatives (4, 4). Due to the divergence of the reviews, I carefully read the paper and made my best efforts to understand the paper and the review comments. This paper proposes to learn a quantization network using a small calibration set given a network trained with the full precision. The combination of AdaQuant, integer programming, and batch-norm tuning makes sense although they do not have substantial novelty. The three components are reasonably tightly-coupled and comprise a complete algorithm. However, the sequential-AdaQuant distracts the main claim of this work significantly. This is probably added during the review process but looks ad-hoc to me. Sequential AdaQuant seems to be effective to improve accuracy, but cannot be applied before the bit allocation was set, which makes it require integer programming no more. Because of this issue, the overall presentation becomes confusing and the argument sometimes sounds unfair (please refer to the last posting by R5.).

In addition, the presentation of this paper could be improved, especially for the details of the integer programming formulation. It is not clear how to define some variables mathematically. The discussion about the size of the calibration set together with the overfitting issue is lacking, and rigorous discussion and analysis would make the paper much stronger. The reviewers are not convinced of the novelty of this paper, and they rather believe that this is an engineering-oriented work. Considering this fact,  the evaluation of this paper is not very comprehensive. The ablation study with respect to the size of the calibration set should be conducted more intensively. The experiment fails to show the benefit of mixed precision quantization effectively and it is limited to presenting the compression ratio in Figure 3. The authors used a small calibration set taken from the training dataset, which looks weird because they claim that the post-training quantization requires only a small "unlabeled" calibration set at the beginning of the abstract; it is more desirable to use arbitrary examples in the same domain.

Despite the interesting aspects, I believe that this paper needs a focus and substantial improvement for publication, and, consequently, recommend rejection.